# SmartCal: A Novel Automated Approach to Classifier Probability Calibration

Mohamed Maher Abdelrahman[1]  Mariam Saadawi[2]  Osama Fayez Oun[2]  Youssef Medhat[2]
Yara Maraey[2]  Abdullah Ibrahim[2]  Radwa ElShawi[1]

[1]Institute of Computer Science, University of Tartu
[2]Innovation hub, GizaSystems

**Abstract**  Accurate probability estimates are crucial in classification, yet widely used calibration methods like Platt and temperature scaling fail to generalize across diverse datasets. We introduce *SmartCal*, an AutoML framework that automatically selects the optimal post-hoc calibration strategy from a pool of 12 methods. Using a large-scale knowledge base of 165 datasets in multiple modalities and 13 classifiers, we show that no single calibrator is universally superior. *SmartCal* employs a meta-model trained on the meta-features of the calibration splits and classifier output to recommend the best calibration method for new tasks. Additionally, Bayesian optimization refines this selection process, outperforming standard baselines and random search. Experiments demonstrate that *SmartCal* systematically improves the calibration over existing approaches such as Beta Calibration and Temperature Scaling. This tool is freely available with a unified interface, simplifying the calibration process for researchers and practitioners.

## 1  Introduction

Accurate probability estimates are central to numerous real-world classification tasks, where reliable confidence scores inform decision-making in domains such as medical diagnosis, fraud detection, and autonomous vehicles (Vaicenavicius et al., 2019). A well-calibrated classifier should produce predicted probabilities that match the true likelihood of each class, enabling users to threshold scores for cost-sensitive classification, adapt to changing class priors, or simply trust the model's reported confidence (Cohen & Goldszmidt, 2004). In high-risk medical settings, an overconfident classifier may endanger patient safety by downplaying a rare but severe condition (Huang, Li, Macheret, Gabriel, & Ohno-Machado, 2020), while underconfident fraud detection systems can lead to unnecessary scrutiny of legitimate transactions (Leevy, Hancock, Khoshgoftaar, & Abdollah Zadeh, 2023).

Miscalibration often occurs when machine learning algorithms, especially complex models, systematically misjudge their own predictive probabilities (Pleiss, Raghavan, Wu, Kleinberg, & Weinberger, 2017). *Post-hoc* calibration techniques address this issue by converting raw classifier outputs into more reliable probabilities. Popular approaches include Platt scaling, isotonic regression, temperature scaling, beta calibration, and binning methods (Silva Filho et al., 2023). Each relies on a pre-trained classifier and fits a separate *calibration map* using dedicated validation data. However, these methods differ in assumptions, complexity, and dataset suitability (Huang et al., 2020; Vaicenavicius et al., 2019), posing a challenge for practitioners uncertain about which technique is best for their scenario (Widmann, Lindsten, & Zachariah, 2019).

Although prior work suggests that calibration effectiveness depends on factors like data distribution and model confidence, no single method consistently outperforms all others (Tao, Zhu, Guo, Dong, & Xu, 2023). In a large-scale study across 165 datasets (160 tabular, 3 image, 2 language) and 12 calibration techniques, we found that performance varied widely with respect to feature

distribution, sample size, and class imbalance. Crucially, no single algorithm dominated across all settings, underscoring the context-dependent nature of calibration.

To address this variability, we propose *SmartCal*, an AutoML-based framework for classifier calibration. Central to *SmartCal* is a meta-model trained on the aforementioned large-scale knowledge base, which leverages dataset meta-features and statistical characteristics about the uncalibrated classifier predictions to identify a short list of potentially optimal calibration algorithms. We further augment this recommendation process with a customized Bayesian optimization step initialized with a prior informed by the meta-model's recommendations. This step is used to fine-tune hyperparameters, more efficiently exploring the calibration space than naive alternatives like random selection. An open-source package provides a unified interface for the included algorithms, easing adoption by practitioners unfamiliar with calibration subtleties.

The contributions of this paper are summarized as follows:

- We introduce *SmartCal*, an AutoML framework for class probability calibration, based on a meta-model trained over a broad collection of datasets of different modalities and base classifiers.

- We provide a comprehensive, user-friendly package integrating twelve diverse post-hoc calibration algorithms under a single interface, accompanied by an open-source code release and reproducible results.[1]

- We present large-scale empirical evidence showing that each calibration method excels under specific conditions, motivating the need for an automated selection strategy.

- We demonstrate that *SmartCal* outperforms random search and baseline calibration methods in multiple experimental setups, validating its effectiveness in real-world scenarios.

## 2 Related Work

*Post-processing* calibration methods adjust probability outputs after a model is trained, aligning predicted probabilities with observed outcomes (Silva Filho et al., 2023). In contrast, some approaches incorporate calibration into the training objective (Kumar, Sarawagi, & Jain, 2018; Thulasidasan, Chennupati, Bilmes, Bhattacharya, & Michalak, 2019), but we do not explore these here.

**Calibration Algorithms..** Non-parametric techniques like Empirical Binning (Naeini, Cooper, & Hauskrecht, 2015) partition predicted scores into bins and replace each bin's output with its empirical positive rate. Isotonic Calibration (Naeini & Cooper, 2016) similarly learns a non-decreasing function for mapping scores to probabilities. Both can be extended to multi-class tasks by applying them class-wise. Parametric methods, such as Platt Scaling (Platt et al., 1999) and Beta Calibration (Kull, Silva Filho, & Flach, 2017), fit transformations from logits to probabilities and can handle multi-class predictions on a per-class basis. Temperature Scaling (Guo, Pleiss, Sun, & Weinberger, 2017) is notably simple, using a scalar multiplier on logits, whereas Vector and Matrix Scaling (Guo et al., 2017) add per-class or matrix-level parameters. Dirichlet Calibration (Kull et al., 2019) generalizes these concepts for richer multi-class mappings, and further specialized methods include Multi-Class Uncertainty Calibration (Patel, Beluch, Yang, Pfeiffer, & Zhang, 2020), Mix-n-Match Calibration (Zhang, Kailkhura, & Han, 2020), Meta-Calibration (Ma & Blaschko, 2021), and Probability Calibration Trees (Leathart, Frank, Holmes, & Pfahringer, 2017). Despite this diversity, no single strategy dominates across all settings, with success often hinging on dataset-specific traits like class imbalance or label distribution (Mortier, Bengs, Hüllermeier, Luca, & Waegeman, 2023).

**Automated Machine Learning (AutoML)..** AutoML frameworks (e.g., Auto-sklearn (Feurer, Eggensperger, Falkner, Lindauer, & Hutter, 2022), TPOT (Olson & Moore, 2016), AutoGluon (Erickson et al., 2020), TabPFN (Hollmann et al., 2025), or LLM-driven pipelines (Hollmann,

---

[1]https://github.com/giza-data-team/SmartCal

Müller, & Hutter, 2023)) have reduced the burden of model selection, hyperparameter tuning, and feature engineering (Singh & Joshi, 2022). However, they rarely incorporate calibration, leaving probability refinement as a manual step despite its importance for reliable decision-making (Cohen & Goldszmidt, 2004). This gap highlights the need for an AutoML-driven calibration approach. Rather than relying on fixed heuristics or user intuition, an automated system can adaptively select the best calibration method based on dataset properties and classifier behavior. Our work addresses this challenge by introducing *SmartCal*.

## 3 Problem Formulation

We consider a classification model $M$ trained on a dataset $D_{\text{train}}$, and a collection of post-hoc calibration algorithms $\mathbf{C} = \{C^{(1)}, C^{(2)}, \dots, C^{(13)}\}$, where each algorithm $C^{(i)}$ is parameterized by hyperparameters $\lambda \in \Lambda$. Calibration algorithms take as input the uncalibrated predictions of $M$ and produce calibrated probability estimates.

Let $D_{\text{cal}}$ be a held-out calibration set used to fit the calibration map, and $D_{\text{test}}$ a disjoint test set used solely for performance evaluation. Given a predefined calibration error metric $E$, our goal is to select the optimal calibration algorithm $C_{\lambda}^{(i)*}$ that yields the best calibrated model on $D_{\text{test}}$. The calibration map $C_{\lambda}$ is trained using $D_{\text{cal}}$, and the selection process is subject to a constrained budget $N$ limiting the number of calibration algorithm and hyperparameter configurations that can be evaluated.

Formally, we aim to solve:

$$C_{\lambda}^{(i)*} \;=\; \arg \min_{C \in \mathbf{C},\, \lambda \in \Lambda} \mathrm{E}\big((C_{\lambda}, M), D_{\text{test}}\big), \tag{1}$$

where $(C_{\lambda}, M)$ denotes the calibrated model obtained by applying the calibrator $C$ with configuration $\lambda$, trained on $D_{\text{cal}}$, to the predictions of $M$.

The proposed *SmartCal* framework automates this selection process by first learning a meta-model that recommends promising calibrators based on dataset meta-features and classifier outputs characteristics, and subsequently refining the hyperparameters of the selected candidates via a constrained Bayesian optimization procedure, subject to the budget $N$.

## 4 Methodology

In this section, we start by motivating the need for automated calibration tools by studying the diversity of best calibration methods (Section 4.1). Then, our proposed approach for automatically selecting and tuning post-hoc calibration algorithms is detailed. The workflow is divided into an *offline* phase (Section 4.2), where a large-scale knowledge base is constructed, and an *online* phase (Section 4.3), where the system exploits this knowledge base to provide calibration recommendations and hyperparameter tuning for new data.

### 4.1 Motivation for SmartCal: Diversity of Best Calibration Methods

In our large-scale study, we gathered the "best" post-hoc calibrator for each (dataset, classifier) pair according to a chosen calibration metric. Figure 1 illustrates the frequency with which each calibration algorithm emerged as the top performer. Although some methods appear more frequently than others, all except one algorithm achieve leading performance in at least a few cases. The observed distribution confirms that **no single calibrator uniformly dominates across all datasets and classifiers**. While `MixAndMatch` and several well-known methods such as `Platt` and `Dirichlet`, frequently emerge as the best choice, even less common calibrators such as `Vector Scaling` and `Isotonic` are occasionally optimal.

This diversity of outcomes suggests that relying on any fixed "default" calibrator would lead to suboptimal calibration for many (dataset, classifier) pairs—highlighting the necessity of automated

selection. This pattern strongly suggests that no single calibration method uniformly outperforms the others across diverse data domains and base classifiers.

Hence, an automated selection approach is necessary to navigate the diverse range of algorithms and identify which one is most likely to excel in a particular scenario.

Second, the distribution itself provides a valuable signal that can be leveraged in the calibrator selection process. The frequencies observed across the large-scale study encode prior information about the relative likelihood of each calibrator succeeding in unseen scenarios. Accordingly, *SmartCal* explicitly incorporates this information as a prior in its Bayesian optimization phase (Sec. 4.3), allowing the meta-model's recommendations to be further informed by empirical evidence from the knowledge base. This design ensures that calibrators with a demonstrated history of strong performance are prioritized during optimization, while still allowing flexibility to explore less frequent but occasionally superior alternatives.

Overall, these findings further validate the motivation behind *SmartCal*: effective calibrator selection requires both data-driven adaptivity (via meta-learning) and principled exploitation of prior empirical knowledge.

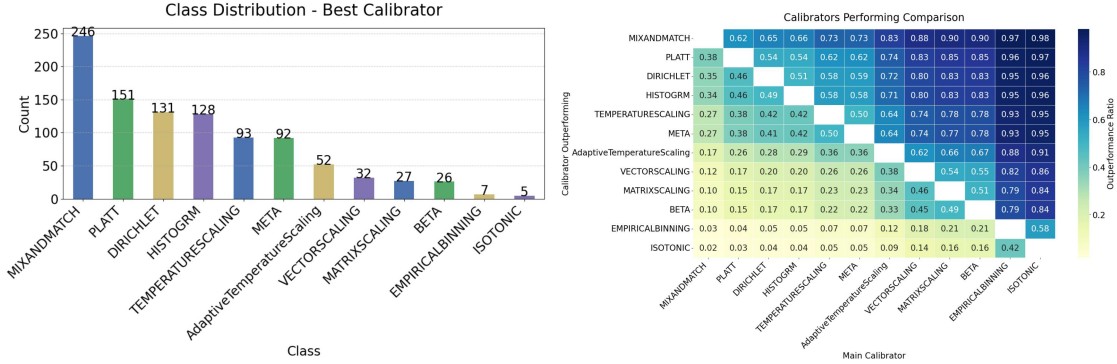

Figure 1: Distribution of top-ranked calibration algorithms demonstrating the absence of a universal "one-fits-all" approach. A histogram of the frequency of each calibrator having the best performance according to ECE in the knowledge base is shown on Left. A heatmap for the ratio of a calibration algorithm outperforms another one in the knowledge base is on the Right.

## 4.2 Offline Phase: Building the Knowledge Base

**Dataset Collection and Preparation**. We gathered an extensive suite of public classification datasets from Kaggle [2], open UCI (Asuncion, Newman, et al., 2007) and OpenML (Vanschoren, Van Rijn, Bischl, & Torgo, 2014), covering tabular, language, and image domains. The datasets were selected, covering a wide range of sizes and count of classes. Each dataset was split into three partitions: a training set (for training the base classifier), a calibration set (for fitting post-hoc calibration algorithms), and a held-out test set (for final performance assessment) with percentages 60%, 20%, and 20%, respectively. Detailed information about these datasets, including their size, number of classes, and feature dimensions, is provided in Appendix A.2.

**Classification Algorithms**. On every dataset, we trained multiple classifiers according to domain relevance. All classification models are detailed in Appendix A.3. These classifiers produced raw (uncalibrated) probability estimates, which serve as the inputs for subsequent calibration.

---

[2] https://www.kaggle.com/

**Calibration Algorithms and Optimization Technique.** We integrated twelve post-hoc calibration algorithms, described in Table 1 with the count of their hyper-parameters, and detailed in Appendix A.4, including widely used methods (Platt scaling, Temperature scaling, Isotonic regression, Beta calibration, various binning approaches) and more specialized algorithms for multi-class calibration (Vector/Matrix/Dirichlet scaling). Each algorithm had its own hyperparameter search space. Using the calibration set for each dataset-classifier pair, we performed a grid search to identify the hyperparameter configurations that optimize calibration performance (Section 4.2.1).

Table 1: Number of numerical and categorical hyperparameters for each calibration algorithm.

| Calibration Algorithm | # Numerical | # Categorical |
|---|---|---|
| Empirical Binning | 1 | 0 |
| Isotonic Calibration | 0 | 0 |
| Beta Calibration | 0 | 1 |
| Temperature Scaling | 3 | 0 |
| Vector Scaling | 2 | 0 |
| Matrix Scaling | 2 | 0 |
| Dirichlet Calibration | 2 | 0 |
| Meta-Calibration | 2 | 1 |
| Platt Scaling | 1 | 1 |
| Histogram-based Calibrator | 1 | 1 |
| Adaptive Temperature Scaling | 4 | 1 |
| Mix-n-Match Calibration | 0 | 2 |

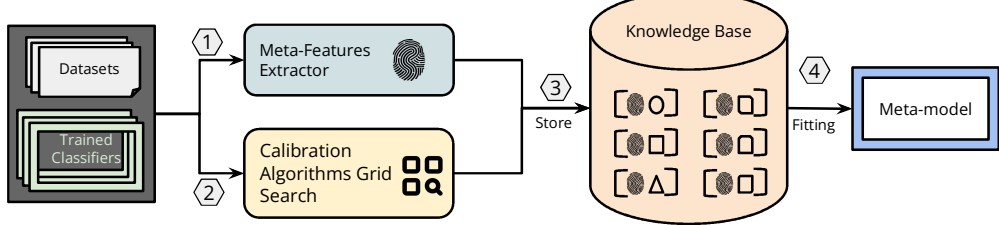

Figure 2: Meta-learning methodology: The meta-model is trained over a constructed knowledge base of the post-hoc calibrators performance over different classification algorithms.

### 4.2.1 Calibration Metrics and Knowledge Base Construction.
After fitting each calibrator's grid-search configuration, we assessed calibration quality using five metrics summarized in Table 2. These metrics capture different perspectives on how well predicted probabilities match observed frequencies (Arrieta-Ibarra, Gujral, Tannen, Tygert, & Xu, 2022).

Table 2: Calibration Metrics Used in the Evaluation.

| Metric | Description |
|---|---|
| **Max Calibration Error (MCE)** | The maximum, over probability bins, of the absolute difference between the average predicted probability and the empirical frequency of the positive class. |
| **Expected Calibration Error (ECE)** | The average absolute difference between predicted probabilities and observed frequencies across all bins, providing a global measure of calibration. |
| **Confidence Calibration Error (Conf ECE)** | A variant of calibration error focusing on the predicted confidence (the maximum predicted probability), often relevant for single-label tasks. |
| **Brier Score** | A proper scoring rule that sums the squared difference between predicted probabilities and actual outcomes. Lower values indicate better calibration and accuracy combined. |
| **Log Loss** | Penalizes overconfident incorrect predictions more heavily. A widely used metric in probabilistic classification. |

For each dataset-classifier pair, we identified the *best* calibration algorithm (and its hyperparameters) according to each of the metrics in Table 2, thus establishing multiple "winners" if using

different metrics. We recorded these results in a *knowledge base* along with the **meta-features** of the calibration set, described next.

**Meta-Feature Extraction**. We extracted a set of meta-features from the calibration set, summarizing dataset properties (e.g., number of classes) and classifier output behaviors (e.g., entropy measures). Table 3 lists the meta-features used, along with the aggregation functions to convert raw measurements into stable scalars.

The rationale for selecting these meta-features is twofold. First, prior research has shown that **dataset characteristics** and **classifier confidence behaviors** influence how well classification and calibration algorithms perform (Vanschoren, 2019; Sayed et al., 2024; Maher & Kull, 2021). Including these dimensions ensures that our meta-model captures variations across both data and model behaviors that are known to affect calibration outcomes.

Second, we adopted a deliberately broad and redundant set of statistical summaries of classifier outputs—including confidence statistics (mean, variance, skewness, kurtosis, min/max), and distribution distances (KL divergence, Jensen-Shannon, Wasserstein, Bhattacharyya) to enable the meta-model to flexibly identify which signals are most informative for recommending calibrators. The use of multiple aggregation statistics provides robustness against noisy or unstable raw measures.

To validate this design choice, we analyzed the feature importance scores of the trained meta-model using SHAP-based explanations. Appendix A.6 presents a detailed feature importance plot, which shows that many of the included meta-features contribute substantially to the meta-model's predictions. This supports our rationale of including a rich set of features spanning both data properties and classifier behaviors.

Table 3: Overview of the meta-features used in knowledge base construction.

| Meta-feature | Details (Aggregations if any) |
|---|---|
| Calibration evaluation metric | |
| Classes count | |
| Num. of instances in calibration set | |
| Class imbalance ratio | |
| Classifier predictions entropy | |
| Classifier confidences | Mean, Median, Std, Var, Skewness, Kurtosis, Min, Max |
| Classification performance | Acc, Micro/Macro F1, Precision, Recall |
| Classifier calibration performance | ECE, MCE, Confidence ECE |
| Wasserstein distance KL Divergence Jensen Shannon Bhattacharyya | between predicted probabilities distributions and actual probability distributions in the calibration set. Aggregated with Mean, Median, Std, Var, Entropy, Skewness, Kurtosis, Min, Max |

**4.2.2 Meta-Model Training.** We partitioned the knowledge base on the dataset level into a training set (80%) and a validation set (20%) in order to learn and evaluate our meta-model. Each entry in the knowledge base indicates which calibration algorithm performed best under specific conditions (dataset meta-features, classifier type, etc.). Instead of fitting a single multi-class model, we opted for a set of one-vs-all Adaboost classifiers, one for each potential "best" calibration method. During inference, each Adaboost model outputs a probability that its associated calibrator is the top performer for a given instance of meta-features. If one or more models produce probabilities exceeding a chosen threshold $T$, we include those corresponding calibrators in the recommended set. Lowering $T$ encourages exploration by allowing more calibrators to be suggested, whereas raising $T$ yields a more selective recommendation.

On the 20% validation split of the knowledge base, the final combined meta-model achieved an F1-score of 82.7% when $T = 0.4$. This threshold provided a practical balance between recommending multiple potentially strong calibrators (exploration) and focusing on the few most likely to be optimal (exploitation).

### 4.3 Online Phase: Algorithm Recommendation and Tuning

In the online phase, our system operates on new datasets or tasks for which it aims to produce high-quality probability estimates via post-hoc calibration. The process illustrated in Figure 3 begins by extracting the same meta-features that were used during the offline phase. These meta-features describe key properties of the new calibration set, such as the shape and distribution of predicted probabilities, class imbalance, and dataset-specific statistics. Once the meta-features are collected, they are fed into the meta-model trained previously, which outputs a ranked list of recommended calibration algorithms. This recommendation is crucial to focus subsequent efforts on the methods that are most likely to perform well in the current setting.

After receiving the ranked list of calibration algorithms, the system devotes a computational budget of $N$ iterations to tuning the top candidates. During this tuning stage, a focused hyperparameter optimization approach—commonly Bayesian optimization (Snoek, Larochelle, & Adams, 2012) is applied. Rather than exhaustively searching over all algorithms and parameter settings, the search is initialized with a prior of the meta-model recommendations. It targets the most promising methods and systematically explores their hyperparameter space. This targeted approach ensures an efficient balance between the exploitation of known good regions in the hyperparameter space and the exploration of less-tested configurations that may yield additional gains. Upon completing the optimization within the allocated time, the system selects the calibration algorithm and hyperparameter set, yielding the best results on a validation subset of the new calibration data. At this point, the chosen calibrator can be applied to the final test set or deployed in production. The resulting calibrated model benefits from the broader knowledge encapsulated in the offline-constructed database, combined with local fine-tuning informed by the characteristics of the current dataset. Hence, the online phase substantially reduces the computational burden compared to a full grid search over the entire calibration algorithms space. At the same time, it leverages the historical experience encoded in the knowledge base, guiding the search for those approaches most likely to perform well.

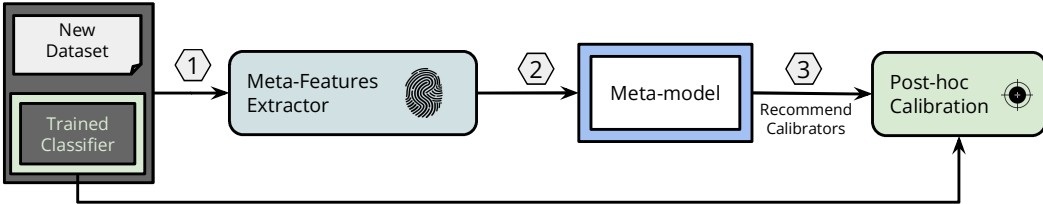

Figure 3: Online pipeline of the proposed methodology. The meta-model will be used to recommend post-hoc calibrators to be used with the trained classifier.

## 5 Empirical Evaluation

This section presents our experimental setup, followed by a thorough evaluation of the proposed *SmartCal* framework from two angles. First, we compare the performance of the meta-model's algorithm recommendation against the random algorithm selection. Second, we contrast the end-to-end *SmartCal* pipeline (which includes meta-model recommendation plus hyperparameter tuning) with a brute-force random search over all calibration algorithms and parameter settings. We include

a statistical analysis of the outcomes based on Wilcoxon signed-rank tests and provide a concluding discussion of key findings.

## 5.1 Experimental Setup

We evaluated *SmartCal* using 30 benchmark datasets spanning tabular, language, and image domains, all distinct from those used to build the knowledge base. Appendix A.2 provides full dataset details, while Appendix A.3 describes the domain-relevant classification models employed. Following the online phase outlined in Section 4.3, we extracted meta-features, obtained the meta-model's recommendations, and performed a constrained hyperparameter search on these recommended algorithms. All experiments were run on a machine with 4 vCPUs, 16 GB of memory, and Red Hat OS (Version 9.4) on an Intel Xeon(R) Gold 6138 CPU @ 2 GHz.

Calibration was evaluated on each dataset's *test* split using ECE, ensuring no overlap with the calibration sets. This protocol reflects true generalization and avoids overfitting to a single partition. We first compared the meta-model's ability to pick the best calibrator against a random selector, highlighting how effectively *SmartCal* identifies the most suitable algorithm for each dataset-classifier pair. Next, we benchmarked the entire *SmartCal* pipeline—meta-model plus hyperparameter tuning—against a random search baseline spanning all calibration algorithms and hyperparameters, as well as two commonly used methods: Temperature scaling and Beta calibration. This approach offers a broad view of *SmartCal*'s practical advantages and limitations.

## 5.2 Comparison of Meta-Model Recommendation vs. Random Algorithm Selection

We compared the meta-model's ability to select the lowest-ECE (best) calibrator on each (dataset, classifier) pair against a random selector. A calibrator is considered selected by the meta-model if its predicted selection probability exceeds a threshold $T > 0.4$. The random selector is configured to select, uniformly at random, the same number of calibrators as selected by the meta-model for each pair. We define the meta-model as correct for a pair if at least one of its selected calibrators matches the best calibrator (i.e., the one achieving the lowest ECE on $D_{\text{test}}$), and analogously for the random selector.

Figure 4 (left) summarizes three possible outcomes for each pair: **Meta-Model Correct** (only the meta-model selects the best calibrator), **Random Correct** (only random selection succeeds), and **Tie** (both select the best calibrator, or neither does). The meta-model proves correct in the majority of cases, showing a marked advantage over random guessing under this selection protocol.

We then examined whether the best calibrator lies within the meta-model's top-$K$ recommendations. For each (dataset, classifier), the meta-model produced $K$ suggestions, while the random selector provided $K$ randomly chosen calibrators. Figure 4 (right plots the fraction of scenarios in which the true best calibrator is included, with $K$ ranging from 1 to 7. The meta-model consistently achieves higher success rates than random selection at all $K$ levels.

We further analyzed the distribution of calibrators selected by the meta-model across all (dataset, classifier) pairs to assess whether it preserves the diversity of successful calibrators observed in the ground truth. As shown in Appendix A.5, the predicted selection frequencies closely resemble the distribution of best calibrators presented in Figure 1. This alignment demonstrates that the meta-model effectively captures the heterogeneity of calibration success patterns, and avoids collapsing onto a narrow subset of algorithms.

## 5.3 End-to-End SmartCal vs. Baselines

We compared our complete *SmartCal* pipeline (meta-model recommendation + hyperparameter tuning) against three baselines on 30 benchmarking datasets (each paired with multiple classifiers), using Expected Calibration Error (ECE) as the key metric. The experiments were repeated thrice with different random seeds of dataset splits, and average results with standard deviation are reported. The baselines include **(i)** Random Search across all calibration algorithms and their

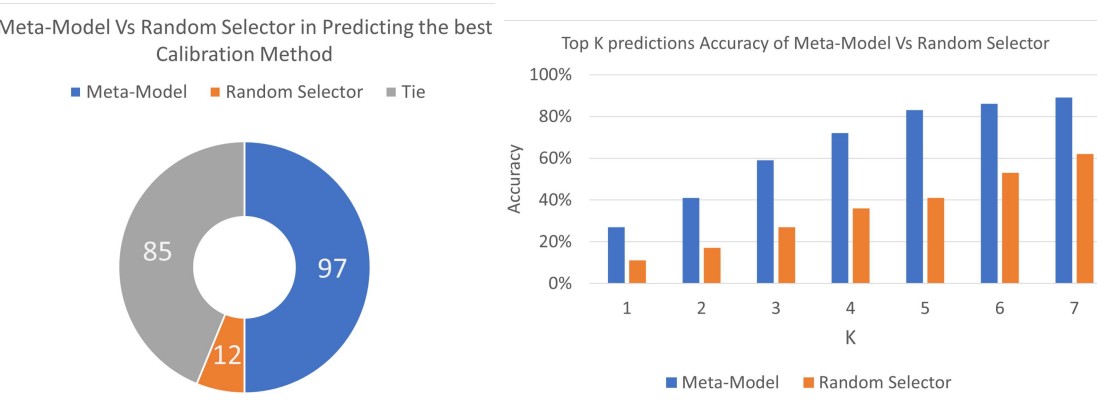

Figure 4: Comparison of meta-model vs. random selector. **Left**: Pie chart illustrating the meta-model predictions' correctness. The meta-model is configured to recommend any calibrator with a predicted probability threshold $T = 0.4$, and random search is configured to select randomly the same number of algorithms as the meta-model. **Right**: Bar chart of top-$K$ inclusion from $K = 1$ to 7. The meta-model consistently outperforms random selection.

hyperparameter spaces (for $N = 10$ and $N = 30$ iterations), **(ii)** Temperature Scaling, and **(iii)** Beta calibration.

Table 4 summarizes the average ECE over all (dataset, classifier) combinations, along with standard deviations. The first two columns show the results for *SmartCal* under two different iteration budgets ($N = 10$ and $N = 30$) followed by Random Search using the same iteration budgets. The last two columns compare the Temperature scaling and Beta calibration methods, each used with its default hyperparameters or minimal tuning.

Table 4: Average ECE and standard deviations across combinations of 30 benchmark datasets and 13 classifiers. Smaller is better.

|          | SmartCal(10) | SmartCal(30) | RS (10)  | RS (30)  | Temp.    | Beta     |
| -------- | ------------ | ------------ | -------- | -------- | -------- | -------- |
| Avg. ECE | 0.0301       | 0.0240       | 0.0382   | 0.0408   | 0.0743   | 0.0405   |
|          | ± 0.0314     | ± 0.0267     | ± 0.0732 | ± 0.0673 | ± 0.0626 | ± 0.0406 |

We observe that *SmartCal* yields the lowest mean ECE compared to all baselines, indicating more reliable calibration on average. Random Search with $N = 10$ performs closer to *SmartCal*, but still exhibits higher variability and slightly worse average ECE. Using Higher random search iterations ($N = 30$) slightly overfits the calibration set, and the calibration error increases more with fewer iterations. Temperature scaling and Beta calibration, though simpler to deploy, show higher average miscalibration and cannot match the adaptiveness of *SmartCal*.

Overall, the proposed framework outperforms both fixed strategies (Temperature scaling and Beta calibration) and unstructured exploration (Random Search). The improvement is especially evident under small iteration budgets, where *SmartCal* makes more efficient use of limited trials. Complete per-dataset performance tables and additional plots are provided in our repository [2] for reference, including detailed values across each classifier, dataset, and calibration metric combination.

### 5.4 Statistical Analysis

We ran a Friedman test followed by a Nemenyi post-hoc analysis (Demšar, 2006) on the average ranks of: *SmartCal* (10 vs. 30 iterations), Random Search (10 vs. 30 iterations), Temperature Scaling

(TempScaling), and Beta Calibration (BetaCal). Figure 5 displays the critical difference (CD) diagram, where lower rank values denote better calibration performance; methods whose intervals do not overlap the *CD* bar differ significantly at the chosen level.

As shown, both *SmartCal* variants achieve lower ranks than the baselines, especially with more iterations (SmartCal30). RandomSearch30 lies between *SmartCal* and the simpler fixed strategies, while RandomSearch10, TempScaling, and BetaCal occupy higher ranks. These findings reinforce our earlier observations that a meta-model–driven approach, combined with time-constrained hyperparameter tuning, yields more reliable calibration than either random or single-method baselines.

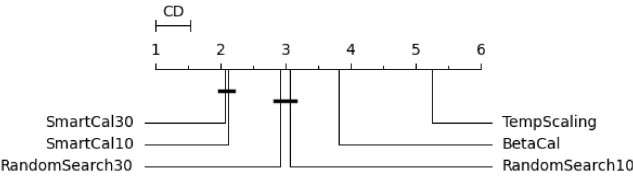

Figure 5: Critical difference diagram from the Friedman/Nemenyi analysis, comparing average ranks of *SmartCal*, Random Search, Temperature Scaling, and Beta Calibration. Lower ranks are better; overlapping intervals indicate no significant difference.

## 6 Conclusion, Limitations and Future Work

We presented *SmartCal*, an automated framework that learns when and how to apply different post-hoc calibration algorithms. By constructing a large-scale knowledge base through extensive experimentation and training a meta-model on dataset meta-features, *SmartCal* offers reliable algorithm recommendations and targeted hyperparameter tuning under a constrained budget. Empirical results showed that this data-driven approach substantially outperforms random search and other baselines, underscoring the value of informed calibrator selection.

Despite these positive outcomes, our work has a few limitations. First, *SmartCal* depends on an offline knowledge base that may not fully represent novel data distributions or emerging calibration algorithms. Second, our current study focuses mainly on standard supervised classification tasks, limiting the generality of our findings to other settings, such as structured outputs or multi-label tasks. Finally, some meta-features may not capture all intricacies of real-world datasets, which can lead to occasional mismatches between recommendations and actual performance.

Future work may extend *SmartCal* to include in-train calibration algorithms, including calibration metrics in the training objective itself, offering a more universal solution. We will also enhance the meta-model interpretability, aiming to clarify why certain calibrators are chosen. Furthermore, integrating incremental updates to the knowledge base would help *SmartCal* adapt to new data and evolving calibration methods. Finally, exploring the robustness to domain shifts and out-of-distribution samples could broaden the real-world applicability of the framework.

## 7 Broader Impact Statement

After careful reflection, the authors have determined that this work presents no notable negative impacts on society or the environment.

**Acknowledgements.** This work was supported by the project "Increasing the knowledge intensity of Ida-Viru entrepreneurship" co-funded by the European Union and the innovation hub at Giza Systems [3].

---

[3]https://gizasystems.com

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
