# OpenReview forum: "SmartCal: A Novel Automated Approach to Classifier Probability Calibration"
_automl.cc/AutoML/2025/Methods_Track — AutoML 2025 Methods Track_

### Official Review · Reviewer_qGcN · 2025-04-30

**Comments To Authors:**

# Review of SmartCal: A Novel Automated Approach to Classifier Probability Calibration

## Summary
Miscalibration is a well-known and important issue in modern machine learning methods. While numerous post-hoc calibration techniques exist, no single method consistently outperforms the others across different datasets, model architectures, and modalities.

This paper undertakes a large-scale empirical study on 172 datasets across multiple modalities, testing 12 state-of-the-art calibration techniques in conjunction with 13 classifiers. Their results confirm the lack of a universally dominant calibration method (see Fig. 1).

To address this, the authors propose *SmartCal*, a meta-learning framework that recommends suitable calibration algorithms for a given task. SmartCal uses dataset-level meta-features (e.g., calibration metric values, class imbalance ratio, etc.) to predict a ranked list of calibration algorithms. (A subset of) these recommended algorithms is then fine-tuned using Bayesian hyperparameter optimization under a constrained computational budget on a validation set. The authors also provide an open-source implementation.

## Relation to Prior Work
- The related work is well-presented and comprehensive. It convincingly motivates the need for AutoML-driven calibration selection, a gap currently underexplored in AutoML according to the authors.
- It might be useful to include a brief discussion of conformal prediction and how it differs from or complements calibration, as the concepts are often linked in practice.
- Line 64 mentions that calibration can sometimes be included in training objectives. If so, is it possible that such approaches offer a more universal solution than any post-hoc technique? Addressing this could reinforce the case for SmartCal's necessity.

## Technical Correctness
Overall, the paper is technically sound, though there are a few important points that merit clarification:

- Eq. (1) in Section 3: The optimization is performed on the calibration set $D_{\text{cal}}$, not the test set. This should be corrected.
- Missing appendix: The appendix is referenced but not available in the main document or the provided repository.
- A key concern is that SmartCal currently depends on the classifier used as an input to the meta-model. Would SmartCal still work if this input is omitted? If yes, that would make it truly standalone and more broadly applicable beyond the 13 classifiers considered.
- Line 45 and 101 suggest the meta-model takes both meta-features and uncalibrated classifier outputs as input. However, Section 4.2.2 implies that only meta-features are used as input; the classifier outputs are calibrated using the recommended method. This discrepancy should be clarified.
- Line 201 claims reduced computational cost relative to grid search, but no supporting data is provided. Quantitative evidence would help validate this claim.
- Line 232–235: The notion of the “correct” classifier is unclear. Is this based on the highest assigned probability? Or simply any calibrator for which $T > t$ (with $t = 0.4$)? Clarification is needed.
- Fig. 4: left shows best vs. random. Though, right, for K=1 is also best vs. random. Yet, there the meta-model seems to perform at least double as good as random (while on the left it is only slightly better)? Perhaps left and right report different metrics, but a more thorough analysis would be beneficial here for the reader to understand strengths and weaknesses of the proposed method.

## Strengths
- The paper addresses a critical gap in AutoML systems: miscalibration is often overlooked, despite its importance.
- The authors clearly define the problem and provide a compelling motivation.
- The dataset collection is diverse in modality, class balance, and size, reflecting real-world classification challenges.
- The study includes up-to-date and varied calibration techniques.

## Weaknesses
- See comments under "Technical Correctness"
- Table 4 is inconsistent: The "beta" column is missing, and the Avg. ECE reported suggests that random search (N = 10) outperforms SmartCal. This contradicts claims in lines 257–266. If the paper is accepted, I strongly recommend revisiting both Table 4 and the surrounding text.
- It is sometimes unclear how many recommendations are considered “correct” or how “correctness” is defined.
- Minor clarity issues:
  - In Section 4.2.2 (lines 166–177), clarify that the data split is done at the dataset level, not instance level.
  - Clarify evaluation methodology when multiple “correct” calibrators may exist.

## Overall Assessment
This is a timely and well-motivated paper that proposes a principled, AutoML-style approach to calibration method selection. The approach is grounded in a rich experimental setup with diverse data. However, the framework’s practical utility and claims would benefit from improved clarity around model inputs, evaluation definitions, and especially result interpretation of Table 4. If these issues are addressed, SmartCal has potential for impact.

## Recommendation
**Borderline Accept** (while the paper holds promise, Table 4 and its surrounding discussion requires attention).

**Review Confidence:**

4

**Review Rating:**

6

---

### Official Review · Reviewer_VXgo · 2025-05-01

**Comments To Authors:**

This paper proposes a system/framework for post-hoc probabilistic calibration named SmartCal. The authors assemble a collection of 172 datasets and 13 classifiers to investigate the performance of 12 calibration algorithms. Of the 12 calibration algorithms, the authors conclude that no single calibration method uniformly outperforms the others. Therefore, they suggest using a meta-model strategy to dynamically select the best calibration strategy for a given model and dataset. The authors proceed by suggesting a set of meta-features (Table 3) to be collected and used to train their meta-model (a collection of regression models) that ranks the list of calibration algorithms. Finally, the system also performs a couple of rounds of Bayesian optimization to tune the performance of the best calibration algorithms, selecting the optimized candidate that achieves best performance on a validation dataset.

(+) This paper is easy to read, and the method is straightforward.
The analysis and investigation presented across 172 datasets, 13 machine learning models, and 12 different calibration algorithms is a good contribution, in my humble opinion. From a method perspective, nothing is novel (the work is all about using well-known machine learning techniques well), but the overall system, findings, and application are relevant, especially considering the model calibration is an important topic commonly overlooked. Additionally, the motivation and related work did a great job contextualizing the contributions, and a system for automated post-hoc calibration is relevant to any machine learning practitioner. This paper could also be a good fit for AutoML application submission.

(-) I have minor comments. For example, the main text does not clarify how you use the N iterations of Bayesian optimization across the multiple calibration algorithm candidates. It may be something simple, such as a round-robin strategy. Yet, in theory, you could have a custom Bayesian optimization system that intelligently selects which candidate algorithm is more appropriate to be fine-tuned next; one way of doing it is to use your meta-model to compute an acquisition function. Answering this question is another research problem, but it would certainly make this paper very strong.

Typos:
Line 94: C must be finite
Table 4 is a bit off in terms of numbers and text and the columns with the text description.

My recommendation is a weak acceptance. It checks all the criteria for a good publication. I don't see any significant flaws, but I wouldn't rank it among the top 20%.

**Review Confidence:**

4

**Review Rating:**

7

---

### Official Review · Reviewer_kmQU · 2025-05-02

**Comments To Authors:**

Title: SmartCal: A Novel Automated Approach to Classifier Probability Calibration

Summary:

This paper proposes the automated selection of calibration methods for (re-)calculating classifier probability. Current approaches use a statically pre-determined calibrator such as temperature scaling that may not result in the high accuracy across different datasets. Through experiments on 172 datasets, the authors demonstrate that different datasets require different calibration methods to ensure high accuracy. The SmartCa approach trains a meta-classifier based on logistic regression to identify a suitable calibrator for different datasets. Experimental evaluations are conducted on 30 datasets, and demonstrate improved accuracy compared to a random selector, and lower average calibration error compared to the temperature scaling method.

Strengths:

The paper is well-written, and the idea of selecting a calibration algorithm is interesting. One of the core contributions of this work is demonstrating that different calibration algorithms perform well on different dataset, that is, there is no one-size-fits-all approach. The SmartCal approach builds on this finding and uses meta-features to predict a suitable calibrator for each dataset.

Weaknesses:
1. Is there any intuition about the 12+85 datasets for which meta model selector did not outperform random?
2. In Table 4, when using a lower budget, random search seems to be outperforming in terms of average ECE. Additionally, the average 3. rank metric has not been explained.
4. Beta scaling values missing in Table 4
5. Why those meta-features specifically? What is the rationale?
6. Figure 4 requires a bit more detailed explanation in the caption. In Figure 4, it would be good to see which calibrators are selected by the meta-selector. It would also be good to visualize the accuracy achieved by the baselines – temperature scaling. This would be similar to the concept of “single best solver” and “virtual best solver” from algorithm selection literature.
On that note, referring to evaluation methodology from algorithm selection literature would be beneficial for this work. Here’s a starting point: Pascal Kerschke, Holger H. Hoos, Frank Neumann, Heike Trautmann; Automated Algorithm Selection: Survey and Perspectives. Evol Comput 2019; 27 (1): 3–45. doi: https://doi.org/10.1162/evco_a_00242
7. A more detailed analysis in Section 4.1 would strengthen this paper, since this is the core premise of your work.

**Review Confidence:**

4

**Review Rating:**

4

---

### Meta-Review · Area_Chair_XkBX · 2025-05-11

**Recommendation:** Accept
**Confidence:** 4

**Metareview:**

The reviews agree with the originality and relevance of the work, they point out a set of weaknesses in the study, many that need to be corrected, but none that affect the relevance of the work. I understand that it is a good work, written in a clear way.

The main pros I identified in the paper and reviews are:

The proposal is original and interesting
The paper is well written and easy to follow and the proposal is straightforward
The experiments confirm that none calibration technique is superior to the others
Address an issue that is overlooked in the AutoML community
A diverse set of datasets was used for the experiments
The experiments included up-to-date calibration techniques
It is possible to replicate the experiments

The main cons are:
The authors should justify their choices and provide a more detailed analysis of the results
A custom Bayesian optimization system could be more appropriated than fine to intelligently selects which candidate algorithm should be fine-tuned
- Address how calibration could be included in the training objectives

Some necessary minor corrections were also pointed out.